# Stem-cell-ubiquitous genes spatiotemporally coordinate division through regulation of stem-cell-specific gene networks

Natalie M. Clark[1,2,7], Eli Buckner[3,8], Adam P. Fisher[1,8], Emily C. Nelson[1,8], Thomas T. Nguyen[1,8], Abigail R. Simmons[4], Maria A. de Luis Balaguer[1], Tiara Butler-Smith[1], Parnell J. Sheldon[1,5], Dominique C. Bergmann[4,6], Cranos M. Williams[3] & Rossangela Sozzani[1,2]*

Stem cells are responsible for generating all of the differentiated cells, tissues, and organs in a multicellular organism and, thus, play a crucial role in cell renewal, regeneration, and organization. A number of stem cell type-specific genes have a known role in stem cell maintenance, identity, and/or division. Yet, how genes expressed across different stem cell types, referred to here as stem-cell-ubiquitous genes, contribute to stem cell regulation is less understood. Here, we find that, in the Arabidopsis root, a stem-cell-ubiquitous gene, TESMIN-LIKE CXC2 (TCX2), controls stem cell division by regulating stem cell-type specific networks. Development of a mathematical model of TCX2 expression allows us to show that TCX2 orchestrates the coordinated division of different stem cell types. Our results highlight that genes expressed across different stem cell types ensure cross-communication among cells, allowing them to divide and develop harmonically together.

[1] Department of Plant and Microbial Biology, North Carolina State University, Raleigh, NC 27695, United States. [2] Biomathematics Graduate Program, North Carolina State University, Raleigh, NC 27695, United States. [3] Department of Electrical and Computer Engineering, North Carolina State University, Raleigh, NC 27695, United States. [4] Department of Biology, Stanford University, Stanford, CA 94305, United States. [5] Department of Biology, Denison University, Granville, OH 43023, United States. [6] Howard Hughes Medical Institute (HHMI), Stanford University, Stanford, CA 94305, United States. [7] Present address: Department of Plant Pathology and Microbiology, Iowa State University, Ames, IA 50011, United States. [8] These authors contributed equally: Eli Buckner, Adam P. Fisher, Emily C. Nelson, Thomas T. Nguyen. *email: ross_sozzani@ncsu.edu

Stem cells asymmetrically divide to replenish the stem cell and produce a daughter cell that will go on to differentiate into a specialized cell type. Various mechanisms have been proposed for how pluripotency is maintained, such as signaling pathways within the stem cell niche (SCN) that restrict differentiation, predetermined lineages which ensure stem cells are continuously formed, and cell plasticity which allows differentiated cells to revert to a stem-like state[1–4]. However, most of the pathways that have been shown to maintain pluripotency use local mechanisms, such as short-range signaling, DNA methylation, and chromatin remodeling, that only act on the dividing cell and/or the directly adjacent cells[5–8]. There are likely other networks, upstream of these local mechanisms, which are global in nature and allow for cross-communication across different cell populations.

The Arabidopsis root provides an excellent model system for uncovering these global regulatory networks. The root SCN, located at the tip of the root, contains several spatially well-defined stem cell populations with different biological roles. At the center of the root SCN is the Quiescent Center (QC), which serves as a relatively mitotically inactive organizing center to maintain the surrounding stem cells in an undifferentiated state. The remaining stem cell initial populations are located adjacent to the QC and asymmetrically divide to form the differentiated root tissues. The Cortex Endodermis Initials (CEI) and the Epidermis/Lateral Root Cap Initials (Epi/LRC) asymmetrically divide twice to form their respective layers. The Columella Stem Cells (CSCs) asymmetrically divide once to form the differentiated columella layer. Finally, the vascular stem cells contain different populations of cells, such as the Xylem initials (Xyl) and the Protophloem (Protophlo), which divide and produce the different vasculature layers[9] (Fig. 1a). As the stem cells asymmetrically divide, the differentiated cells are displaced towards the root, providing a temporal developmental axis where older cells are more shootward and younger cells are more rootward. Crucially, the movement of cells in the root is constrained due to cell walls, and symplastic cell-to-cell signals travel via the plasmodesmata, which are small channels in the cell walls[10]. This lack of cell movement coupled with well-defined marker lines that label specific cell populations[11,12] allows us to study stem cell identity, division, and maintenance in an isolated environment.

Here, we identify genes expressed specifically (in one stem cell type) and ubiquitously (in all stem cell types) that control stem cell division and maintenance in the Arabidopsis root. We first transcriptionally profile the individual stem cells using spatially well-defined GFP marker lines and find that nearly half of the stem cell-enriched genes are expressed in only one stem cell type, while the other half are expressed in multiple cell types. We next use Gene Regulatory Network (GRN) inference to predict that there are not only stem-cell-specific gene networks but also an upstream network that regulates all of the different stem cells. Given that most known mechanisms for maintaining stem cell identity and plasticity are local in nature, we focus on identifying genes expressed in all the stem cells (hereinafter referred to as stem-cell-ubiquitous genes) that regulate aspects of stem cell maintenance. Using our network prediction, we find that TESMIN-LIKE CXC 2 (TCX2), a member of the family of CHC proteins which are homologues of components of the DREAM cell-cycle regulatory complex in animals[13], is a key regulator of stem cell division. Further, using Ordinary Differential Equation (ODE) modeling, we show that we can use the dynamics of TCX2 expression to predict the timing of stem cell division. Our results provide evidence that genes that participate in global regulatory pathways which span many, different cell types are important for controlling stem cell division and maintenance.

## Results

**Determining stem-cell-specific and -ubiquitous genes.** To understand how and whether stem-cell-ubiquitous genes contribute to cell identity, maintenance, and/or division, we performed gene expression analysis of the stem cells in the Arabidopsis root, as this offers a tractable system given its 3-dimensional radial symmetry and temporal information encoded along its longitudinal axis. To this end, seven root stem cell markers (Fig. 1a), as well as a non-stem cell control (i.e., a population of cells from the root meristem excluding most of the stem cells), were used to identify stem cell-enriched genes, and among those, stem-cell-ubiquitous and stem-cell-specific genes, as it has been shown that there is a correlation between expression levels and functionality in specific cell types[1,14] (Supplementary Fig. 1, see Methods). Notably, we found that the expression profiles of our markers together with known stem cell genes agree with their known expression domains, supporting that our transcriptional profiles are specific to each stem cell population (Supplementary Fig. 1). To measure transcriptional differences between the stem cells and the non-stem cells, we next performed a Principal Component Analysis (PCA). Looking at the top three principal components (50.6% of the variation in the data), the PCA shows that the non-stem cell samples (red) are distant from all of the stem cell populations, suggesting that stem cells have a different transcriptional signature than the non-stem cells (Fig. 1b). Accordingly, when we performed differential expression analysis on these data, we found that 9266 (28% of genes) are significantly enriched (q < 0.06 and fold-change > 2) in at least one stem cell population compared to the non-stem cells and considered these genes the stem cell-enriched genes (see Methods and Supplementary Data 1). Thus, this approach allowed us to identify core stem cell genes, as functionally important genes are often enriched in the specific cell populations they control[1,14].

While the PCA gives us a general idea of how many genes are cell-specific vs cell-ubiquitous, it reduces the dimensionality of the problem to the three largest components of variance. Consequently, we would expect some genes are differentially enriched across all of the stem cell populations. Indeed, when we performed differential expression analysis on the 9266 stem cell-enriched genes (see Methods), we found that 2018 genes (21.8% of the stem cell-enriched genes, hereinafter referred to as the stem-cell-ubiquitous genes) are enriched in at least four of the six unique stem cell types, with 569 of these 2018 (6.1% of the stem cell-enriched genes) enriched in five or six cell types (Fig. 1c). Moreover, as each stem cell population clusters independently from the others in the PCA, we identified 7248 genes (78.2% of the stem cell-enriched genes), hereinafter referred to as the stem-cell-specific genes, enriched in three or less stem cell types, with 4331 of those 7248 genes (46.7% of the stem cell-enriched genes) enriched in only one stem cell type. This suggests that each specific stem cell type has its own, unique transcriptional signature.

**Root stem cell regulatory network prediction.** Given the separation between stem-cell-ubiquitous genes and stem-cell-specific genes, we next wanted to know if these two groups of genes have seemingly separated functions or, for example, if stem-cell-ubiquitous genes modulate stem-cell-specific gene expression to orchestrate coordinated processes between different cell types. To test the latter hypothesis, in which stem-cell-specific genes are important for regulating cell-type-specific aspects (e.g., cell identity), but are regulated by stem-cell-ubiquitous genes so that stem cell maintenance and divisions are tightly coordinated, we used Gene Regulatory Network (GRN) inference and

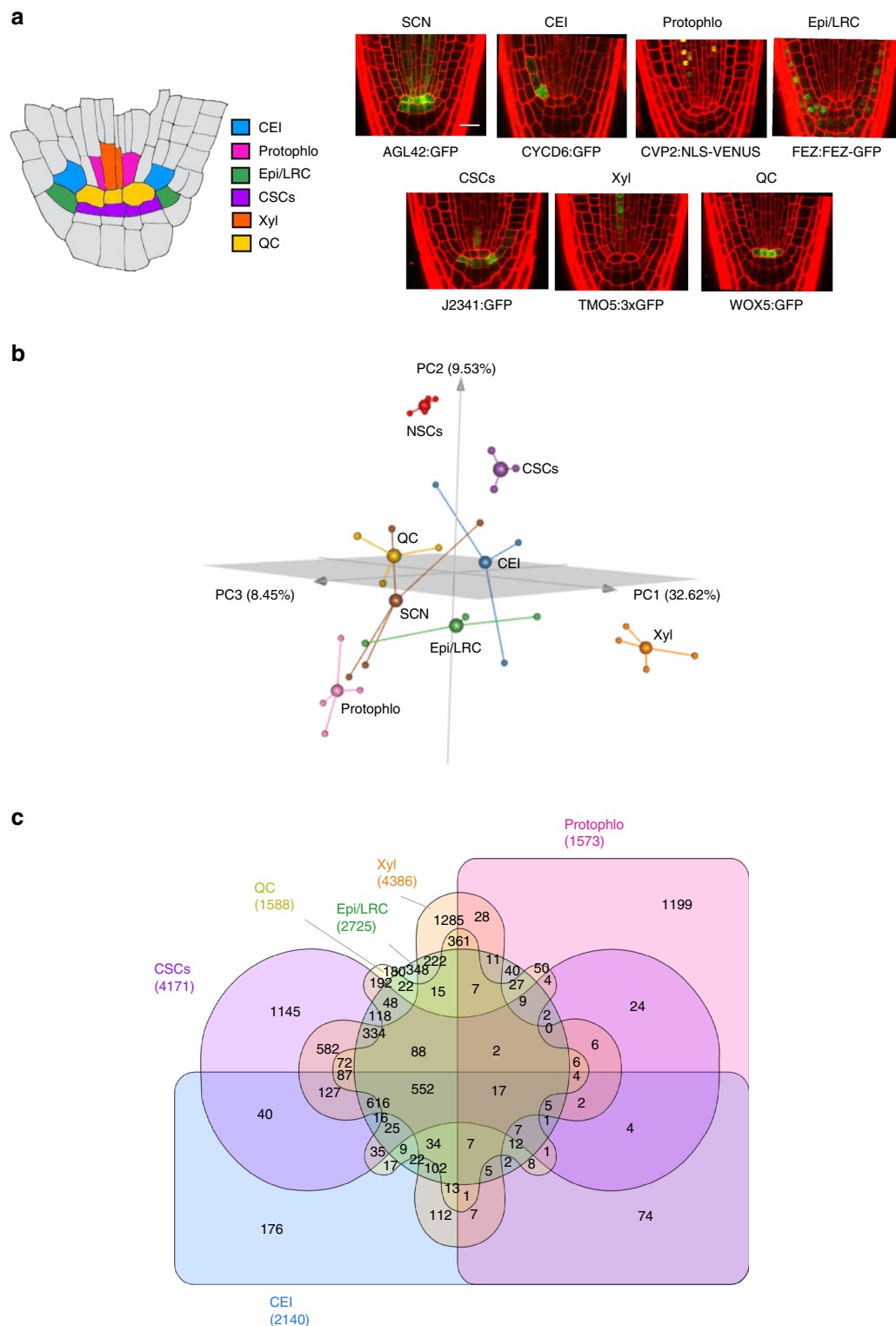

**Fig. 1** Distribution of genes within the Arabidopsis root stem cell niche. **a** (left) Schematic of the Arabidopsis root stem cell niche. CEI cortex endodermis initials (blue), Protophlo protophloem (pink), Epi/LRC epidermis/lateral root cap initials (green), CSCs columella stem cells (purple), Xyl xylem initials (orange), QC quiescent center (yellow). (left) GFP marker lines used to transcriptionally profile stem cells. SCN stem cell niche; Scale bar = 20 μm. **b** 3D principal component analysis (PCA) of the stem cell transcriptional profiles. The x, y, and z axis represent the three largest sources of variation (i.e., three largest principal components) of the dataset. Small spheres are biological replicates, large spheres are centroids. Red—Non-stem cells (NSCs); Brown—SCN; Blue—CEI; Pink—Protophlo; Green—Epi/LRC; Purple—CSCs; Orange—Xyl; Yellow—QC; **c** Distribution of the 9266 stem cell-enriched genes across the stem cell niche. Enrichment criteria are q-value < 0.05 (from PoissonSeq) and fold-change in expression >2.

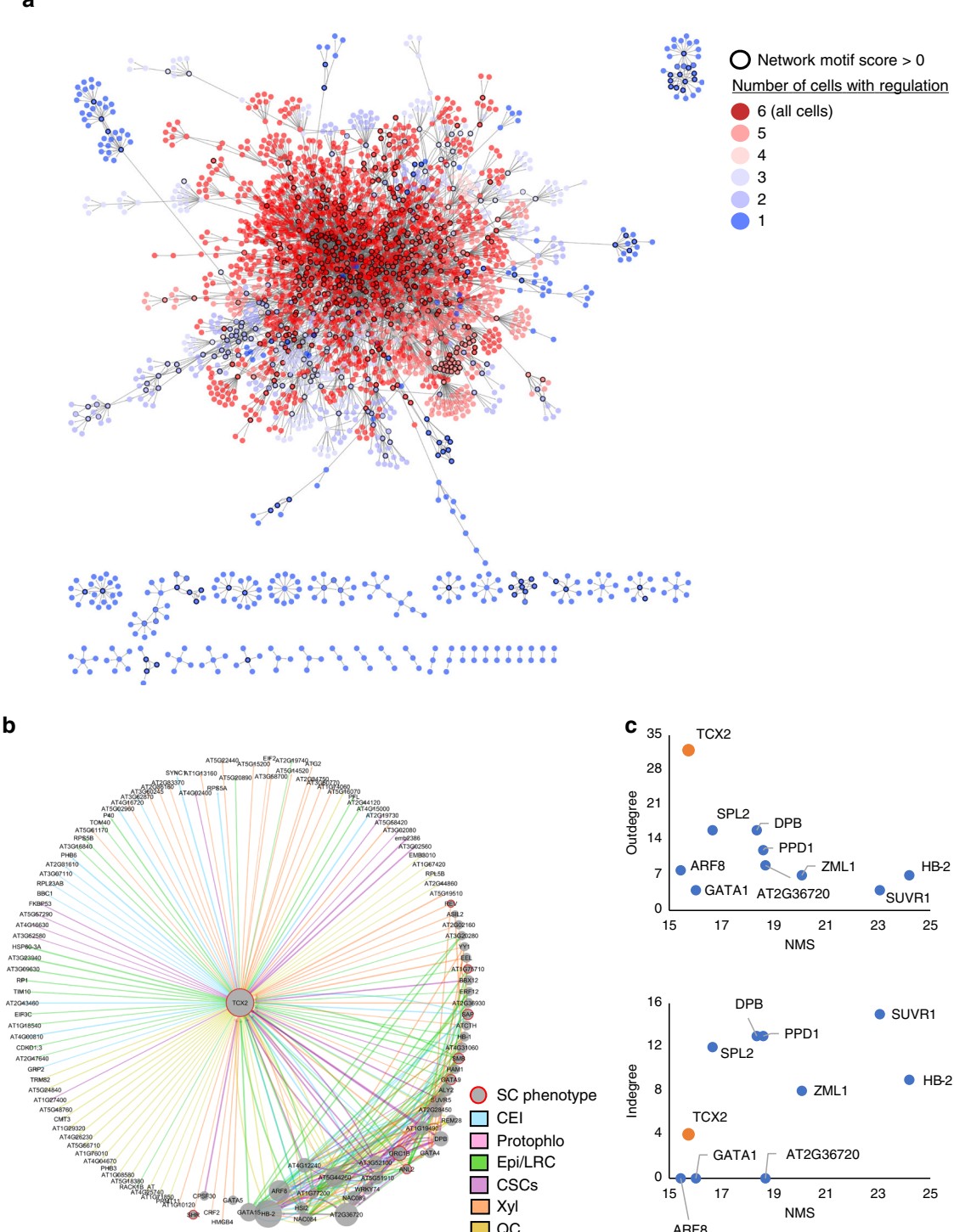

**Fig. 2** Gene regulatory network (GRN) of the stem cell-enriched genes. **a** Inferred GRN of 2982 out of the 9266 stem cell-enriched genes. Genes are colored based on the number of genes in which they are enriched, with red genes (>3 enriched cells) considered stem-cell-ubiquitous and blue genes (≤3 enriched cells) considered stem-cell-specific. Black outlines represent hub genes which have a network motif score (NMS) > 0. **b** First-neighbor GRN of TCX2. Gene size represents the NMS score. Red borders represent the genes which have a known stem cell (SC) phenotype. Edge colors represent the cell in which the regulation is inferred. Blue—CEI; Pink—Protophlo; Green—Epi/LRC; Purple—CSCs; Orange—Xyl; Yellow—QC. **c** Outdegree (top plot) and indegree (bottom plot) vs NMS score of the genes with the top 10 NMS scores in **a**. TCX2 is highlighted in orange.

predicted the relationships between all 9266 genes enriched in the stem cells. We developed a machine-learning, regression tree-based approach to infer dynamic networks from steady state, replicate data (see Methods). Our inferred GRN found regulations among 2982 (32.2%) of the stem cell-enriched genes and

predicted that the stem-cell-ubiquitous (red) genes are located in the center of the network, which represents the beginning of the regulatory cascade, and are highly connected to each other (Fig. 2a). Meanwhile, the cell-specific (blue) genes mostly regulate each other within the same cell type and are located on the

outside of the network, downstream of the cell-ubiquitous genes (Fig. 2a). This suggests that the cell-ubiquitous genes are potentially involved in coordinating processes between different stem cells through the regulation of cell-specific genes.

We next wanted to identify if the most biologically important genes in the network were stem-cell-specific, stem-cell-ubiquitous, or both, as most results in animals assume that core TFs must be expressed in a cell-specific manner[1]. To predict biological significance, we developed a Network Motif Score (NMS) to quantify the number of times each gene appears in certain network motifs, such a feedback and feed-forward loops (see Methods). These motifs were chosen as they were significantly enriched in our biological network versus a random network of the same size and have been shown to often contain genes that have important biological functions[15–17] (Supplementary Fig. 2). In our inferred GRN, we found that 737 (24.7%) of the 2982 genes have an NMS > 0, meaning they appear in at least one of the network motifs. To validate the NMS, we found that 22 known stem cell regulators had scores in the top 50% of genes, with 10 of those 22 (45.5%) in the top 25% of genes, supporting that high NMS scores are correlated with stem cell function (Supplementary Data 2). Further, 510 (69.2%) and 217 (31.8%) of these genes are stem-cell-ubiquitous (four or more enriched stem cells, red) and stem-cell-specific (three or less enriched stem cells, blue), respectively (Fig. 2a). Given that more cell-ubiquitous genes have higher importance scores in our dataset, we focused our downstream analysis on identifying a stem-cell-ubiquitous gene with characteristics of a functionally important regulator.

**TCX2 controls stem cell division and identity**. When we began to examine the stem-cell-ubiquitous regulators, we found that TESMIN-LIKE CXC 2 (TCX2, also known as SOL2), a known homologue of the LIN54 DNA-binding component of the mammalian DREAM complex, which regulates the cell cycle and the transition from cell quiescence to proliferation[13,18,19], had the ninth highest NMS (top 1.2% of genes). This suggests that TCX2 could have an important role across all of the stem cells. To further support the biological significance of TCX2, we examined the subnetwork of its first neighbors (i.e., genes predicted to be either directly upstream or downstream of TCX2). We found that TCX2 is enriched in five out of the six stem cell types and predicted to regulate at least one gene in all of those cell types, supporting that TCX2 could be a stem-cell-ubiquitous regulator that controls stem-cell-specific core genes (Fig. 2b). In addition, when compared to the genes with the top 10 NMS, TCX2 has the highest outdegree (number of edges going out) and relatively low indegree (number of edges coming in), suggesting that TCX2 could orchestrate coordinated stem cell division as suggested by the function of its mammalian homologue[13,18,19].

If TCX2 is indeed a key regulator for stem cell maintenance and division, we would expect that a change in its expression would cause a developmental phenotype related to these aspects. To test this hypothesis, we obtained two knockdown (tcx2-1, tcx2-2) and one knockout (tcx2-3) mutant of TCX2, which all show similar phenotypes (Fig. 3a, Supplementary Fig. 3). Importantly, we observed in tcx2-3 an overall disorganization of the stem cells, including aberrant divisions in the Quiescent Center (QC), columella, endodermis, pericycle, and xylem cells (Fig. 3a). Additionally, tcx2-3 mutants showed longer roots due to a higher number of meristematic cells, suggesting higher cell proliferation (Fig. 3a, Supplementary Fig. 3). Notably, similar phenotypes related to cell divisions have been observed also in the stomatal lineage of tcx2 sol1 double mutants[13]. To further investigate TCX2's role in stem cell division, we crossed the cell division (G2/M phase) marker CYCB1;1:CYCB1;1-GFP[20,21] into the tcx2 mutant and performed temporal tracking of the GFP signal over time. We first found that average CYCB1;1 expression was higher in the tcx2 mutant compared to WT. Second, we separated cells expressing CYCB1;1 into three categories: low, intermediate, and high expression. We found that significantly more cells in the tcx2 mutant have high CYCB1;1 expression, while significantly fewer cells have low CYCB1;1 expression. Finally, we calculated the number of consecutive timepoints each cell showed CYCB1;1 expression. We found that significantly fewer cells in the tcx2 mutant had two consecutive timepoints with CYCB1;1 expression (Supplementary Fig. 4). All of these alterations in CYCB1;1 expression in the tcx2 mutant suggest that reduction of TCX2 expression correlates with more actively dividing cells. Taken together, these results suggest that TCX2, as a stem-cell-ubiquitous gene, regulates stem cell divisions across different stem cell populations.

We hypothesized that TCX2 controls stem cell division by regulating important, stem-cell-type-specific genes. Notably, all of our stem cell markers, in addition to being expressed in only one stem cell type, are known to have functions in stem cell regulation[22–26]. Thus, we crossed the marker lines for the Quiescent Center (QC; WOX5:GFP), Cortex Endodermis Initials (CEI; CYCD6:GFP), Epidermis/Lateral Root Cap Initials (Epi/LRC; FEZ:FEZ-GFP), and Xylem Initials (Xyl; TMO5:3xGFP) (Fig. 1a) into the tcx2-2 and tcx2-3-mutant alleles (Fig. 3b). Compared to WT, in a tcx2 mutant the expression pattern of these markers is expanded. Specifically, the QC marker expands into the CEI, the CEI marker expands into the endodermis and cortex layers, the Epi/LRC marker expands into the Columella Stem Cells (CSCs), and the Xyl marker expands into the procambial cells (Fig. 3b). This suggests that, in the absence of TCX2, coordination of stem cell division and identity is misregulated through an unknown mechanism.

When we examined the predicted upstream regulators and downstream targets of TCX2, we found that 75% are predicted to be cell-specific (expressed in ≤3 stem cell types), suggesting that TCX2 could be regulated and itself regulate targets in a cell type-specific manner. (Supplementary Data 3). Thus, to identify additional cell-specific regulators as well as targets of TCX2, we obtained mutants of the transcription factors (TFs) predicted to be TCX2's first neighbors (i.e., directly upstream or downstream) that also had high NMS scores (Fig. 3c, Supplementary Data 3). Two of the genes, SHORTROOT (SHR), and SOMBRERO (SMB) have phenotypes in the stem cells of their loss-of-function mutants, while the loss-of-function mutant of STERILE APETALA (SAP) is homozygous sterile[23,25,27–29]. Additionally, a quadruple mutant of REVOLUTA (REV) together with three other xylem regulators results in missing xylem layers[30]. Further, we obtained loss-of-function mutants of GATA TRANSCRIPTION FACTOR 9 (GATA9), AT1G75710, ORIGIN OF REPLICATION COMPLEX 1B (ORC1B), ANTHOCYANINLESS 2 (ANL2), and REPRODUCTIVE MERISTEM 28 (REM28), which showed root stem cell phenotypes (Fig. 3c, Supplementary Fig. 5). We were able to validate that TCX2 was differentially expressed in gata9, at1g75710, rev, orc1b, and anl2 mutants using qPCR as well as in the SHR overexpression line[23]. Further, we performed FACS coupled with RNA-Seq on the 4 marker lines (WOX5:GFP, CYCD6:GFP, FEZ:FEZ-GFP, and TMO5:3xGFP) that we crossed into the tcx2 mutant to determine the effect of TCX2 on its predicted downstream stem-cell-specific targets. In addition, we performed RNA-Seq on tissue from the stem cell area of the tcx2 mutant (Supplementary Data 5). Using these data, we were able to validate that 77.8% of the predicted direct targets of TCX2 are differentially expressed in the tcx2-mutant stem cells. Further, 41.5% of these edges are predicted in the correct cell type, and of those edges predicted in the correct cell type that had a predicted

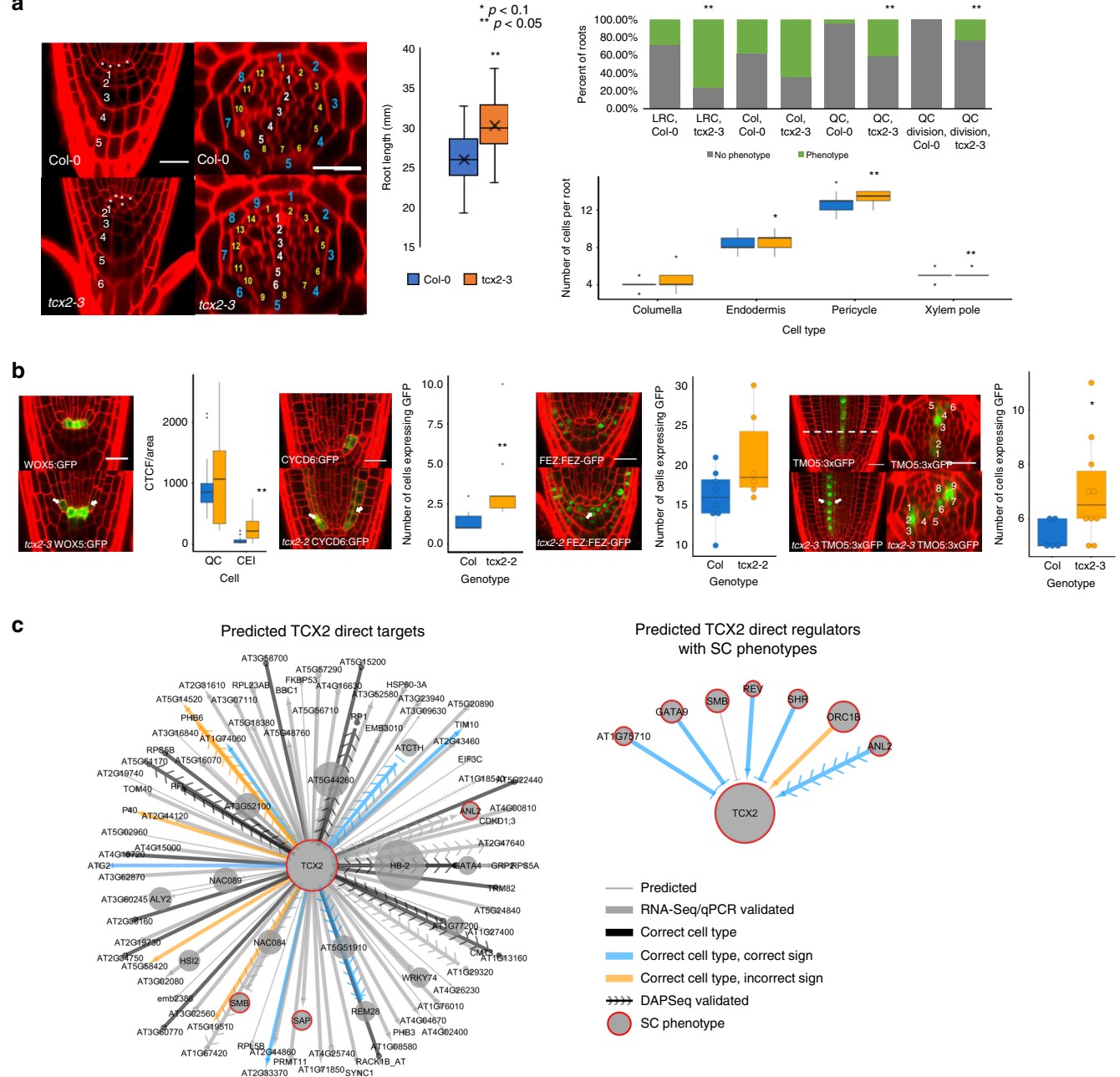

**Fig. 3** TCX2 controls stem cell division through cell-specific regulators and targets. **a** (Left) Medial longitudinal (left) and radial (right) sections of 5-day-old WT (top) and *tcx2*-mutant (bottom) plants. In medial longitudinal sections, asterisk labels QC cells and numbers denote columella cell files. In radial sections, white numbers denote xylem cells, yellow pericycle, and blue endodermis. Scale bar = 20 μm. (Middle) Length of 7-day-old WT (blue, n = 18) and *tcx2* mutant (orange, n = 18) roots. Center line represents median; box bounds represent 25th and 75th percentiles; whiskers represent minimum and maximum values. Black X represents the mean. (Right) Quantification of stem cell phenotypes (top plot) and number of cell files (bottom plot) in 5-day old WT (n = 21) and *tcx2* mutant (n = 17) roots. In top plot, green denotes the percent of roots that showed a phenotype, while gray denotes the percent of roots that did not show a phenotype. LRC, Col (Columella), QC refers to a cell disorganization phenotype in those cells. Asterisk denotes p < 0.1, Double asterisks denote p < 0.05, Wilcoxon test. **b** (Left) Medial longitudinal sections of 5-day-old WOX5:GFP (left), CYCD6:GFP (second from left), FEZ:FEZ-GFP (third from left), and TMO5:3xGFP (right) in WT (top) and *tcx2*-mutant (bottom) plants. For TMO5:3xGFP, a radial section (middle) was taken at the location of the white, dashed line. (Right) Quantification of GFP in WT (blue, n = 35, n = 8, n = 5 for CYCD6, FEZ, TMO5) and *tcx2*-mutant (orange, n = 25, n = 6, n = 10 for CYCD6, FEZ, TMO5) plants. Center line represents median; box bounds represent 25th and 75th percentiles; whiskers represent minimum and maximum excluding outliers. Black dots represent outliers. Colored dots represent individual data points. Asterisk denotes p < 0.1, double asterisks denote p < 0.05, Wilcoxon test. **c** (Left) Predicted direct targets of TCX2 and (right) predicted upstream regulators of TCX2 with stem cell (SC) phenotypes. Gene size represents the NMS score. Red borders represent the genes that have a known SC phenotype. Arrows represent predicted activation, bars repression, and circles no predicted sign. Thick edges were validated using qPCR/RNA-Seq. Black edges were predicted in the correct cell type but did not have a predicted sign. Blue edges have the correct cell type and correct sign, while orange edges have the correct cell type but the incorrect sign. Arrows with chevrons are DAPSeq validated. Source data are provided as a Source Data file.

sign, 58.3% of the edge signs are correctly predicted (slightly better than randomly assigning edge signs, which would have a 50% rate of success). To validate some of the direct interactions between TCX2 and its downstream targets, we mined a published DAP-Seq dataset from Arabidopsis leaves[31] and were able to confirm that TCX2 can directly bind 15.1% of its predicted direct targets (Fig. 3c, Supplementary Data 4). Overall, these results suggest TCX2 orchestrates coordinated stem cell divisions through stem-cell-specific regulatory cascades.

**TCX2 network changes over time to regulate cell division.** Given that most of the validated upstream regulators of TCX2 are stem-cell-type-specific (Supplementary Data 3), we propose that these stem-cell-specific regulators modulate the dynamics of TCX2 expression in individual cell types. In turn, changes in TCX2 dynamics correlate with changes in expression of its downstream targets (Fig. 3c, d). Thus, we hypothesized that different dynamics of TCX2 in specific stem cells, as well as changes in TCX2 expression, could be used to predict when each stem cell population divides.

If TCX2 expression is dynamically changing over time in a cell-specific manner, we would predict that the TCX2 GRN also changes temporally. Specifically, we could expect that TCX2 differentially regulates its targets in specific cell types at certain times depending on its expression levels. Thus, to determine if the TCX2 regulatory network changes over time, we first selected 176 genes of interest that were differentially expressed in the *tcx2* root tip sample (Supplementary Data 5) as well as enriched in the stem cells, as these are most likely to be downstream of TCX2 across different stem cell populations. We inferred GRNs using a time course of the root meristem that is stem-cell-enriched (hereinafter referred to as the stem cell time course, see Methods) to predict one network per time point (every 8 h from 4 to 6 days). We found that genes in the first-neighbor network of TCX2 have different predicted regulations depending on the time point. Specifically, most of the regulations to and from TCX2 are predicted to occur between 4 days (4D) and 5 days (5D), which is the developmental time at which many stem cell divisions take place[23] (Supplementary Fig. 6). Thus, since our gene expression data suggest that loss of TCX2 function correlates with an increase in stem cell division, we hypothesized that most of the TCX2-regulated stem cell division is occurring between 4D 16H and 5D when the largest decrease in TCX2 expression occurs (Supplementary Fig. 6).

To test how these time- and cell-specific GRNs affect TCX2 expression and therefore cell division, we built a mechanistic model of the GRNs predicted every 8 h from 4D to 5D (see Methods and Supplementary Information). We used our stem cell time course to determine the stem-cell-specific networks at each time point and constructed equations for each gene in the network (Fig. 4a, Supplementary Fig. 7). Unlike our GRN, which only predicts the regulations in each cell at each time point, our Ordinary Differential Equation (ODE) model converts the network prediction into a quantitative model of gene expression. Thus, this model allowed us to quantify how TCX2 dynamics change over time and to correlate significant changes in expression with cell division. Our model included the possibility of some of the proteins moving between cell types, as this is a known local signaling/cell-to-cell communication mechanism[28,32]. Specifically, we used scanning Fluorescence Correlation Spectroscopy (Scanning FCS) and observed that TCX2 does not move between cells, thus suggesting a cell-autonomous function, while observed movement of WOX5[33] and CRF2/TMO3 between cells is in line with a non-cell-autonomous function (Supplementary Fig. 8). As our sensitivity analysis predicted that the oligomeric state of TCX2 in the Xyl,

diffusion coefficient of WOX5 from the CEI to the QC, and diffusion coefficient of WOX5 from the QC to the Xyl were three of the most important parameters in the model, we experimentally determined these parameters (Supplementary Fig. 8, Supplementary Data 6). Given that our network and time course data predict that TCX2-mediated cell division is tightly coordinated and controlled between 4D 16H and 5D, we wanted to ensure that we accurately measured TCX2 dynamics in this time period to produce the best predictive model of stem cell division. To this end, we quantified the expression of the TCX2:TCX2-YFP marker in different stem cells every 2 h from 4D 18H to 4D 22H (hereinafter referred to as the YFP tracking data) (Fig. 4b, see Methods). We then used the average expression of TCX2 in each cell at each time point to estimate parameters in our model (Supplementary Data 7). The result of this model is thus a spatiotemporal map of the expression dynamics of TCX2 and its predicted first neighbors. Given that TCX2 expression has previously been shown to disappear 1–2 h before stomatal division[13], we reasoned that we could use our model of TCX2 expression to predict when stem cell division occurs in the root.

Our model predicts that there is a significant (fold-change >1.5) increase in TCX2 expression specifically in the QC and Xyl between 4D 8H and 4D 16H. After this time, our model predicts that the expression of TCX2 in the QC does not significantly decrease and is significantly higher than in all of the actively dividing stem cells (Fig. 4c, Supplementary Data 8). Given that the QC is relatively mitotically inactive, this suggests that high levels of TCX2 correlate with a lack of QC division. This prediction is supported by our YFP tracking data, which shows that half of the QC cell clusters have either relatively constant or increased TCX2 expression between 4D 16H an 5D (Supplementary Fig. 9). Meanwhile, TCX2 expression is predicted to significantly decrease between 4D 16H and 5D in both the Xyl and CSCs, suggesting that these cells divide during this time. This prediction is also supported by our YFP tracking data showing that the majority of Xyl and CSCs cells have low TCX2 expression after 4D 20H (Supplementary Fig. 9). In contrast, the CEI and Epi/LRC show only a modest decrease in TCX2 expression between 4D 16H and 5D. This could be due to only some of these cells dividing at that time, as our YFP tracking data show a large amount of variation in TCX2 expression in these cell populations (Supplementary Fig. 9). Taken together, our model and experimental data both suggest that TCX2 not only initiates the division of the actively dividing stem cells, but it also inhibits the division of the QC during the same timeframe, through an unknown mechanism. Further, our results allow us to narrow the timing of TCX2-induced stem cell division to a 4-hour window, between 4D 20H and 5D.

## Discussion

Here, we unraveled the communication between stem-cell-specific and stem-cell-ubiquitous networks in the Arabidopsis root through a combination of transcriptomic profiling, GRN inference, biological validation, and mathematical modeling. Our stem cell transcriptional profile revealed that there is both a stem-cell-specific profile that likely provides the foundation for stem cell identity networks as well as a stem-cell-ubiquitous profile that encodes the unique properties shared by all stem cells, such as their ability to asymmetrically divide. Further, our GRN inference predicted that these stem-cell-specific and stem-cell-ubiquitous networks are connected, with the stem-cell-ubiquitous regulators potentially coordinating the downstream stem-cell-specific mechanisms.

Using our network motif score, we identified TCX2 as an important stem-cell-ubiquitous gene that regulates stem cell

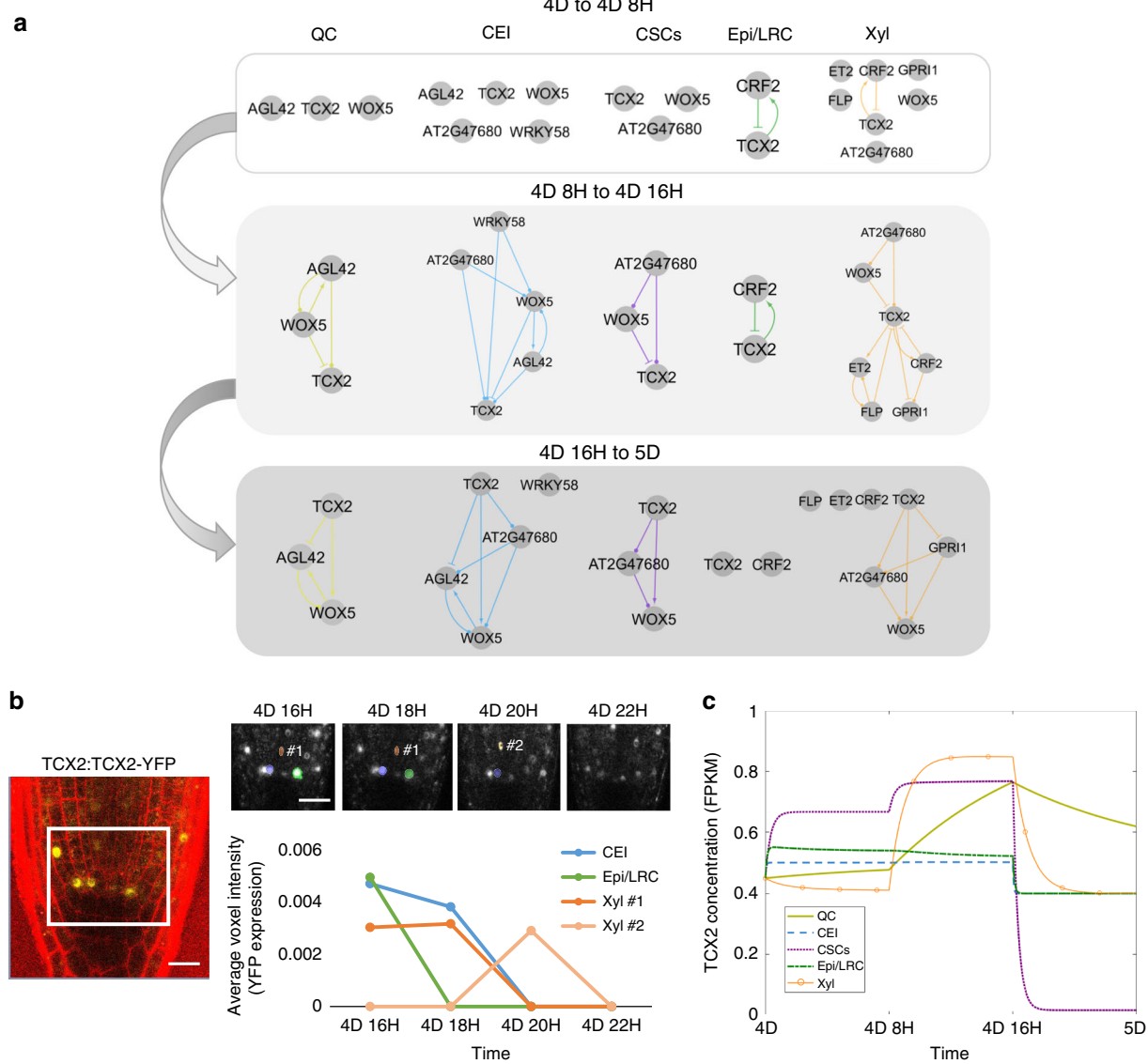

**Fig. 4** Mathematical modeling of TCX2 network predicts timing of cell division. **a** TCX2 first-neighbor TF networks predicted using RTP-STAR on the stem cell time course for 4 day (4D) to 4 days 8 h (4D 8H) (top), 4D 8H to 4D 16H (middle), and 4D 16H to 5D (bottom). Networks are separated based on the cell type the genes are expressed in: QC (yellow), CEI (blue), CSCs (purple), Epi/LRC (green), Xyl (orange). Arrows represent predicted activation, bars inferred repression, and circles no inferred sign. **b** (left) Representative image of TCX2:TCX2-YFP at 4D 16H. White box represents the stem cell niche were cells were tracked over time. Scale bar = 20 μm. (right, top) YFP-positive cells tracked every 2 h from 4D 16H (left) to 4D 20H (right). Stem cells that were tracked are marked in blue (CEI), green (Epi/LRC), and orange (Xyl). Two Xyl cells were tracked, #1 and #2. All of these four stem cells had no measurable YFP expression at 4D 22H. (right,bottom) Quantification of YFP expression in tracked cells. **c** ODE model prediction of cell-specific TCX2 expression from 4D to 5D. FPKM: fragments per kilobase per million mapped reads.

division by coordinating stem-cell-specific regulatory networks. We validated that TCX2 regulates stem-cell-specific genes through transcriptionally profiling some of the stem cell populations in the *tcx2* mutant, supporting that stem-cell-ubiquitous and stem-cell-specific genes work together to coordinate cell division. Specifically, we were able to validate that 77.8% of the predicted direct targets of TCX2 are differentially expressed in the *tcx2* mutant, 44.5% are predicted in the correct cell type, 58.3% have the correct sign, and 15.1% are directly bound by TCX2. Our results showed that most of the stem cell markers are mis-expressed in other stem cells in the *tcx2* mutant (Fig. 3), suggesting that either TCX2 could affect the stem-cell-specific localization of some of these genes or that their cell identity is delayed.

We showed that *tcx2* mutants have additional cell divisions in all stem cell populations, misexpression of known stem-cell-

specific marker genes, and higher expression of the cell-cycle marker CYCB1;1. Further, our ODE model of the TCX2 GRN illustrated that we can use TCX2 expression to predict the timing of stem cell division. Specifically, our model and TCX2:TCX2-YFP tracking support that a drop in TCX2 expression in most of the stem cell populations between 4D 16H and 5D correlates with stereotypical stem cell division. In contrast, TCX2 levels are relatively stable during this time in the relatively mitotically inactive QC. This is supported by our stem-cell-specific profiling of the *tcx2* mutant, which shows that many cell-cycle genes, including members of the CYCLIN and CYCLIN DEPENDENT KINASE families, are differentially expressed in different stem cell types (Supplementary Data 5). TCX2 is a DNA-binding CHC-domain protein, whose sole homologue in animals, LIN54, is a dedicated member of the DREAM complex[13]. The

best-characterized animal DREAM complexes repress cell-cycle gene expression similar to what we observed in the Arabidopsis root. In plants, the number of CHC-domain proteins has expanded, and there is evidence for both redundancy and specialization in the family[13,34]. For example, in the leaf stomatal lineage TCX2 and its closest homologue are redundantly required for division and lineage progression in each of several different cell types[13]. The relatively mild division phenotype we observe in *tcx2* root stem cells suggests that other members of the CHC family, perhaps associated with different formulations of the DREAM complex, also contribute to division regulation. It remains an open question whether TCX2 acts globally to coordinate cell divisions across stem cell types, or if it has separate cell-autonomous functions in each of these different cell types.

Taken together, our results provide evidence that cell-ubiquitous genes and global signaling mechanisms are important for maintaining stem cell identity and plasticity.

## Methods

**Lines used in this study**. A list of T-DNA insertion lines used in this study is provided in Supplementary Data 9. All T-DNA insertion lines were obtained from the Arabidopsis Biological Resource Center (ABRC: https://abrc.osu.edu/). The marker lines used in this study are described as follows: WOX5:GFP[22], CYCD6: GFP[23], J2341:GFP[35], FEZ:FEZ-GFP[25], TMO5:3xGFP[26], CVP2:NLS-VENUS[36], AGL42:GFP[37]. The TCX2:TCX2-YFP translation fusion is described in ref. [13], the WOX5:WOX5-GFP translational fusion is described in ref. [38], the CYCB1;1: CYCB1;1-GFP translational fusion is described in ref. [21], and the TMO3:TMO3-GFP translational fusion is described in ref. [39].

**Stem cell transcriptional profile**. Three to four biological replicates were collected for each marker line. For each biological replicate, 250–500 mg of seed were wet sterilized using 50% bleach, 10% Tween and water and stratified at 4 °C for 2 days. Seeds were plated on 1× MS, 1% sucrose plates with Nitex mesh and grown under long day conditions (16 h light/8 h dark) at 22 °C for 5 days. Approximately 1–2 mm of the root tip was cut and protoplasted using a solution containing cellulase and pectolyase. GFP-positive cells were collected using a Dako Cytomation MoFlo using the gates provided in Supplementary Fig. 10. For the non-stem cell control, the GFP-negative cells from the AGL42:GFP line were collected. RNA was extracted using the Qiagen RNEasy Micro Kit. Libraries were prepared using the SMART-Seq v4 Ultra Low Input RNA Kit for Sequencing and Low Input Library Preparation Kit (Clontech)[35]. Libraries were sequenced on an Illumina HiSeq 2500 with 100 bp single-end reads. Reads were mapped and FPKM (fragments per kilobase per million mapped reads) values were obtained using Bowtie, Tuxedo, and Rsubread[40]. Data are available on Gene Expression Omnibus (GEO: https://www.ncbi.nlm.nih.gov/geo/), accession #GSE92804.

Differential expression analysis was performed using PoissonSeq[40,41]. First, stem cell-enriched genes were identified as being enriched (q-value < 0.06 and fold-change > 2) in any one stem cell population compared to the non-stem cell control (q-value cutoff of 0.06 was chosen since one of the marker genes, WOX5, had q-value 0.058). Then, genes were classified as enriched in each stem cell type if they had fold-change > 2 (enrichment criteria set based on the marker genes) in that stem cell type versus all other stem cell types. If genes were equally expressed in more than one stem cell type, they were considered enriched in multiple stem cell types. All differentially expressed genes are reported in Supplementary Data 1. The Venn diagram in Fig. 1c displaying the proportions of genes enriched in each stem cell was constructed using InteractiVenn[42] (http://www.interactivenn.net/).

**Gene regulatory network inference**. The Regression Tree Pipeline for Spatial, Temporal, and Replicate data (RTP-STAR) was used for all network inference. The pipeline consists of three parts: spatial clustering using the $k$-means method[43], network inference using GENIE3[44], and edge sign (positive/negative) inference using the first order Markov method[11]. An earlier version of this pipeline was used to infer GRNs of root hair development[45]. This pipeline is implemented in MATLAB and available from https://github.com/nmclark2/RTP-STAR.

For the SCN GRN (Fig. 2), networks were inferred for each stem cell separately (resulting in six networks, one for each stem cell) and then combined to form the final network. For the stem-cell-specific networks, only the genes enriched in that specific stem cell were used in the network inference. If genes were enriched in multiple stem cells, they were included in all of those individual stem cell networks (e.g., TCX2, which is enriched in all of the stem cells except Protophlo, was included in five of the six stem cell networks). Genes were first clustered using the mean expression of each gene in each stem cell. Then, network inference was performed using GENIE3 on only the replicates from that specific stem cell and the SCN marker (e.g., for the QC-enriched cells, only the WOX5:GFP and AGL42:GFP replicates were used). After network inference, the number of edges in the network

was trimmed based on the proportion of transcription factors (more transcription factors = more edges kept). Finally, the sign of the edge was determined using a previously published time-course dataset of Arabidopsis root stem cells collected from 3-day to 7-day-old plants[11].

For the time point-specific GRNs (Fig. 4 and Supplementary Fig. 6), we used genes DE in the *tcx2*-mutant root tissue sample and enriched in the stem cells. Clustering was performed as for the SCN GRN using mean gene expression in each stem cell. Network inference was performed using the biological replicates from each time point from our stem cell time course collected every 8 h from 4-days to 6-days-old (see Stem cell time-course section for more details). Edge sign was determined using this same time course but using mean expression in all of the timepoints. One network was built using the biological replicates for each time point and then combined. In Fig. 4, the stem cell transcriptomic data was used to determine the stem cell type of each edge.

Due to the pseudo-random nature of $k$-means clustering (i.e., the first clustering step is always random), 100 different clustering configurations (numiter = 100 in RTP-STAR parameters) were used for network inference. For the stem cell transcriptional network, edges that appeared in at least 1/3 of the 100 different networks (maxprop = 1/3 in RTP-STAR parameters) were retained in the final network as this cutoff resulted in a scale-free network. This parameter was set to edges that appeared in at least 45% of the 100 different networks (maxprop = 0.45 in RTP-STAR parameters) for the time point-specific GRNs.

All parameters used to infer these networks in RTP-STAR are included in Supplementary Data 10. All files used to perform GRN inference are available on figshare (see Data Availability section). All network visualization was performed using Cytoscape (http://cytoscape.org/).

**Network Motif Score (NMS)**. Five different motifs were used to calculate the NMS namely feed-forward loops, feedback loops, diamond, bi-fan, and multilayer motifs[15–17] (Supplementary Fig. 2). All motifs were significantly enriched in the SCN GRN relative to a randomly generated network of the same size. First, the number of times a gene appeared in each motif was counted using the Net-MatchStar app[46] in Cytoscape. Then, the counts were normalized to a scale from 0 to 1 and summed to calculate the NMS for each gene. The most functionally important genes are those that have high NMS scores.

**Biological validation**. Confocal imaging was performed on a Zeiss LSM 710. Cell walls were counterstained using propidium iodide (PI). Corrected Total Cell Fluorescence (CTCF) was calculated to determine the intensity of cells expressing a fluorescently tagged protein. To complete these measurements, the confocal settings (gain, digital offset, laser percentage) were left constant for the entirety of the experiment. Imaging software (ImageJ) was used to measure the CTCF, which is defined as (Integrated density of GFP)/(Area of selected cells * Mean fluorescence of background) where background is a region of the root with no GFP[47]. The CTCF was divided by the area of the cells (CTCF/area) before performing statistics to account for different numbers of cells selected in each image. When counting cells with GFP expression, a local auto threshold using the Phansalkar method was applied in ImageJ to the GFP channel before counting.

For qPCR, total RNA was isolated from approximately 2 mm of 5-day-old Col-0, *gata9-1*, *gata9-2*, *at1g75710-1*, *at1g75710-2*, *rev-5*, *orc1b-1*, *orc1b-2*, *anl2-2* and *anl2-3*, root tips using the RNeasy Micro Kit (Qiagen). qPCR was performed with SYBR green (Invitrogen) using a 7500 Fast Real-Time PCR system (Applied Biosystems) with 40 cycles. Data were analyzed using the ΔΔCt (cycle threshold) method and normalized to the expression of the reference gene UBIQUITIN10 (UBQ10). qPCR was performed on two technical replicates of two to three independent RNA samples (biological replicates). Differential expression was defined as a $p < 0.05$ using a z-test with a known mean of 1 and standard deviation of 0.17 (based on the Col-0 sample). Primers used for qPCR are provided in Supplementary Data 11. SHR regulation of TCX2 was validated using data from[20].

**Stem-cell-specific transcriptional profiling in *tcx2* mutant**. Three biological replicates were collected for WOX5:GFP, CYCD6:GFP, FEZ:FEZ-GFP, and TMO5:3xGFP crossed into the *tcx2-2* or *tcx2-3*-mutant background. Seedlings were grown and roots were collected as described above. Libraries were sequenced on an Illumina HiSeq 2500 with 100 bp single-end reads. Reads were mapped and FPKM (fragments per kilobase per million mapped reads) values were obtained using Bowtie, Tuxedo, and Rsubread[40]. Differential expression analysis was performed using PoissonSeq[40,41]. To account for differences in library size between the stem cell transcriptional profile and the TCX2 cell-specific transcriptional profile, library sizes were normalized before differential expression was performed. A differential expression cutoff of q < 0.05 and fold-change > 2 was set based on the cutoff for the stem cell transcriptional profile. All differentially expressed genes are reported in Supplementary Data 1.

For the *tcx2-3* transcriptional profile, total RNA was isolated from approximately 2 mm of 5-day-old Col-0 and *tcx2-3* root tips using the RNeasy Micro Kit. cDNA synthesis and amplification were performed using the NEBNext Ultra II RNA Library Prep Kit for Illumina. Libraries were sequenced on an Illumina HiSeq 2500 with 100 bp single-end reads. Reads were mapped and differential expression was calculated as above, except the differential expression

criteria were chosen as $q < 0.5$ and fold-change $> 1.5$ based on the values for TCX2, which was assumed to be differentially expressed in its own mutant background.

All differentially expressed genes are reported in Supplementary Data 5. Data for both the stem cell-type-specific profiling and root tip profiling are available on GEO, accession #GSE123984.

**TCX2:TCX2-YFP and CYCB1;1:CYCB1;1-GFP tracking**. Confocal images of the TCX2:TCX2-YFP, CYCB1;1:CYCB1;1-GFP, and CYCB1;1:CYCB1;1-GFP x *tcx2* lines were obtained by imaging roots submerged in agar every 2 h. A MATLAB-based image analysis software (https://github.com/edbuckne/BioVision_Tracker) was used to detect, segment, and track individual cells expressing YFP/GFP in 3D time-course fluorescence microscopy images[48]. The average voxel intensity, which is a proxy for YFP/GFP expression, was measured as the average voxel value within the set of voxels describing a segmented cell.

**Scanning fluorescence correlation spectroscopy**. Image acquisition for Scanning FCS was performed on a Zeiss LSM 880 confocal microscope. For Number and Brightness (N&B) on the TCX2:TCX2-YFP and 35S:YFP lines, the parameters were set as follows: image size of 256 × 256 pixels, pixel dwell time of 8.19 μs, and pixel size of 100 nm. The 35S:YFP line was used to calculate the monomer brightness and cursor size as described in[28,49]. For Pair Correlation Function (pCF) on the 35S:GFP, TCX2:TCX2-YFP and TMO3:TMO3-GFP lines, the parameters were set as follows: image size of 32 × 1 pixels, pixel dwell time of 8.19 μs, and pixel size between 100–500 nm. The movement index (MI) of the 35S:GFP line was used as a positive control. All analysis was performed in the SimFCS software as described in[28,49].

**Stem cell time course**. Two to three biological replicates were collected for each time point. For each biological replicate, 100–250 mg of PET111:GFP seed were wet sterilized using 50% bleach, 10% Tween and water and stratified at 4 °C for 2 days. Seeds were plated on 1× MS, 1% sucrose plates with Nitex mesh and grown under long day conditions (16 hr light/8 hr dark) at 22 °C for 4 days, 4 days 8 h, 4 days 16 h, 5 days, 5 days 8 h, 5 days 16 h, and 6 days. Roots were collected at the same time of day for all samples to minimize circadian effects. GFP-negative cells were collected as PET111:GFP marks only the differentiated columella, so collecting the surrounding GFP-negative cells results in a population of mostly stem cells. Protoplasting, cell sorting, RNA extraction, and library preparation were performed as described above. Libraries were sequenced on an Illumina HiSeq 2500 with 100 bp single-end reads. Reads were mapped and FPKM (fragments per kilobase per million mapped reads) values were obtained using Bowtie, Tuxedo, and Rsubread[40]. Data are available on Gene Expression Omnibus (GEO: https://www.ncbi.nlm.nih.gov/geo/), accession #GSE131988.

**Ordinary Differential Equation (ODE) modeling**. ODE equations were constructed based on the GRNs shown in Fig. 4a. One set of equations was built for each gene in each cell type. The equations changed at 4D 8H and 4D 16H to account for the changes in the predicted network (as shown in Fig. 4a). If a sign was not predicted in the network, it was assumed that the regulation was positive (activation) in the model. A schematic showing the location of genes, and what proteins can move between cell types, is presented in Supplementary Fig. 7. All equations are provided in Supplementary Methods.

A sensitivity analysis was performed using the total Sobol index[28,50,51]. Sensitive parameters were defined as having a significantly higher ($p < 0.05$) total Sobol index than the control parameter using a Wilcoxon Test with Steel-Dwass for multiple comparisons. (Supplementary Data 6) The sensitive diffusion coefficients and oligomeric states were experimentally measured using scanning FCS. The remainder of the parameters were estimated either directly from the stem cell time course or by using simulated annealing[52] on the stem cell time course. For simulated annealing, Latin hypercube sampling was used to sample the parameter space for a total of 50 sets of initial parameter estimates. Each set of initial estimates was fit to the residual function using simulated annealing with least squares (simulannealbnd function in MATLAB) for 5 min (total runtime = 250 min for 50 sets of initial estimates). The average of the 10 parameter values with the lowest error was used in the final model simulation. All parameter values, and how they were estimated, are reported in Supplementary Data 7. All MATLAB files used for the ODE model are available on figshare (see Data Availability section).

**Statistics**. For all confocal phenotyping and RICS analyses, a two-tailed Wilcoxon test (for one comparison) or Steel-Dwass with control (for multiple comparisons) was used as some of the data did not follow a normal distribution. All exact *p*-values, test statistics, and sample sizes are included in Source Data.

**Reporting summary**. Further information on research design is available in the Nature Research Reporting Summary linked to this article.

## Data availability

Sequence data that support the findings of this study have been deposited in GEO with the primary accession codes #GSE98204, GSE123984, and GSE131988. The raw images and data used for GRN inference and biological validation are available in figshare [https://doi.org/10.6084/m9.figshare.c.4539071.] The source data underlying Fig. 3 and Supplementary Figs. 3, 4, 5, 8, and 9 are provided as a Source Data file.

## Code availability

MATLAB code for the ODE model that support the findings of this study are available in figshare [https://doi.org/10.6084/m9.figshare.c.4539071.] RTP-STAR is available from https://github.com/nmclark2/RTP-STAR. YFP/GFP tracking software is available from https://github.com/edbuckne/BioVision_Tracker.

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

## Acknowledgements

We thank Christian Hardtke and Antia Rodriguez-Villalon for providing the CVP2:NLS-VENUS line. We thank Rüdiger Simon and Barbara Berckmans for providing the WOX5: WOX5-GFP line. We thank Irena Brglez for her assistance with media preparation. We thank Sarah Schuett and the Flow Cytometry and Cell Sorting Laboratory at North Carolina State University (NCSU) for their assistance with cell sorting. Images in this manuscript were generated using the instruments and services at the Cellular and Molecular Imaging Facility (CMIF) at NCSU. Next-generation sequencing was performed by the Genomic Sciences Laboratory (GSL) at NCSU. This work was supported by an NSF GRF (DGE-1252376) awarded to N.M.C. and A.P.F. E.B. is supported by a GAAN Fellowship in Molecular Biotechnology (grant #P200A160061). A.R.S. was supported by the Donald Kennedy Fellowship and NIH graduate training grant NIH5T32GM007276 to Stanford University. P.J.S. was supported by an Integrated Molecular Plant Systems Research Experience for Undergraduates (IMPS REU) grant awarded to NCSU. D.C.B. is an investigator of the Howard Hughes Medical Institute. Research in the RS lab was funded by an NSF CAREER grant (MCB-1453130) and the NC Agricultural & Life Sciences Research Foundation in the College of Agricultural and Life Sciences at NC State University.

## Author contributions

N.M.C. and R.S. conceptualized the study and designed the experiments. N.M.C. and A.P.F. performed transcriptional profiling. N.M.C. and M.A.d.L.B. performed differential expression analysis. N.M.C., E.C.N., T.T.N., T.B.S., and P.J.S. performed biological validation. N.M.C., E.C.N., T.T.N., and P.J.S. collected confocal images. N.M.C. constructed and analyzed the mathematical model. A.R.S. and D.C.B. contributed the TCX2: TCX2-YFP translational fusion. E.B. and C.M.W. analyzed the YFP tracking data. N.M.C. made all main and supplementary figures. N.M.C. and R.S. wrote the paper, and all co-authors edited the paper.

## Competing interests

The authors declare no competing interests.
