## [Peer Review File · Nature Communications]

Reviewers' comments:

Reviewer #1 (Remarks to the Author):

Clarke et al. present a transcriptome analysis of the known stem cell types in the root apical meristem and a comparison to “non-stem cells” of the root. They used these data to define genes that are expressed in many stem cell types (stem cell-ubiquitous genes) and those that are expressed in just one or a few stem cell types (stem cell-specific genes). Gene regulatory network inference was then used to derive gene networks for each cell type, allowing them to ask whether stem-cell ubiquitous factors might regulate genes in a stem cell-specific manner and thus contribute to the activity of the stem cell pool. Analysis of the network characteristics identified TCX2 as a stem cell-ubiquitous factor with many predicted downstream targets and relatively few predicted upstream regulators, which could be expected for a “hub” gene in the network. Genetic analysis showed that *tcx2* mutants indeed had defects in cell division control in most of the root stem cell types, and that some of the genes proposed to be first network neighbors of TCX2 showed alterations in expression in the *tcx2* mutants. Finally, the authors performed a time course transcriptome analysis of the root tip and used these data, along with TCX2 imaging, to generate mathematical models of TCX2 expression and network connections over time in each stem cell type. This led to the accurate predictions of the timing of each stem cell, based on a reduction in TCX2 levels. Overall, this is an outstanding study that defines new roles for TCX2 and hypothesizes mechanisms for its action in regulating stem cell division. In my opinion, this manuscript needs only minor revisions prior to publication. Below I point out issues that I think should be addressed, in decreasing order of importance:

1. Regarding the TCX2 regulatory network, the authors should at least point out that these are inferred and thus remain to be tested directly (in subsequent studies) by determining TCX2 binding in each cell type as well as by transcriptome profiling of each cell type in WT and *tcx2* mutants. Along this line, it would be nice to see some discussion of why the expression changes of TCX2 first-neighbors are so subtle in the *tcx2* mutant (Extended Data Figure 4B).
2. I also wonder whether the non-stem cell control (NSC; Figure 1B) used in this study was the best choice. This control is the transcriptome of GFP negative cells from the AGL42:GFP sort. Given that AGL42 only appears to be expressed in only 3 of the 6 stem cell types (Fig 1 A and Extended data Fig 1B), the NSC still contains three other stem cell types. Also, the nature of the NSC should be described in the main text so that readers understand what it is.
3. The timeframe of the “stem cell time course” and associated models and data in Figure 4 need some clarification. The time course is stated as being from 4D to 6D - is this days post-germination? I assume that this actually just represents any two diurnal cycles, since the stem cells are dividing continuously, rather than only once after 4 to 6 days. Is it possible to describe this as a 48 hour time course (e.g. with time zero being at sunrise on day 1) so as to be more intuitive? For example, the statement on Lines 179-181 that “most regulation of TCX occurs between 4-5 days, as does important stem cell divisions” is confusing.
4. In Figure 2A, is it possible to highlight some of the pertinent connections between stem cell ubiquitous and stem cell-specific networks? One can't really see much of what is going on here in that regard due to the high density. Also, there are no visible arrows, bars, or open circles as referenced in the legend.
5. There are a number of grammatical mistakes in the manuscript that need to be corrected.
6. The Extended Data tables should be in spreadsheet format so that they can be more easily used by

readers.

Reviewer #2 (Remarks to the Author):

In the manuscript NCOMMS-18-38553-T titled 'Stem-cell-ubiquitous genes spatiotemporally coordinate division through regulation of stem-cell-specific gene networks', authors infer the gene regulatory networks based on the cell type specific transcriptome profiling for major cell type precursors in the Arabidopsis root meristem and find that the predicted core stem cell regulators acting upstream of specific cell type determinants tend to be stem cell ubiquitous. Among those core regulators, authors report that TCX2 coordinates the stem cell division by dynamically changing its expression domain in different cell type precursors.

Overall, the manuscript well integrates the GRN inference and biological validation, which will contribute to advancing the plant stem cell research. However, several key aspects need to be fully addressed to make the manuscript scientifically thorough.

Major comments:

1. Authors conclude that core stem cell regulators in plants tend to be ubiquitous whereas those in animals are cell-type specific (lines 100-101). This is based on the study by D'Alessio et al. (2015), in which authors searched for core regulators that likely determine fates of different cell types in human, and found regulators that conferred retinal pigment epithelial cell identity when they were overexpressed. In my opinion, the work by D'Alessio et al. (2015) was performed in different contexts from this manuscript. D'Alessio et al.'s aimed to find cell type determinants whereas this manuscript aimed to find coordinators of cell type determinants. Thus, contrasting and generalizing cell type expression patterns of stem cell regulators in plants and animals based on these two studies seems an overstatement.

2. This manuscript does not explain clearly about the significance of coordinating the timing of stem cell division in the stem cell maintenance of the root meristem. It would be helpful if authors provide the long-term consequence of perturbing TCX2 or some other relevant data.

3. Lines 49-51: the PCA shows that the non-stem cell samples (red) are distant (using the Euclidean distance metric) from all of the stem cell populations, suggesting that the stem cells have a different transcriptional signature than the non-stem cells (Figure 1B).

Comment: It is not clear to me by looking at the PCA plot. It would be helpful if the coordinates of each sample or the Euclidean distance matrix is provided.

4. From line 175-

The experimental approaches used to infer the time-dependent networks of TCX2 are not clear.

Describe the experiments in more detail in materials and methods.

5. From line 206:

It is not clear to me how authors specified times for time series experiments of transcript profiling and quantification of TCX2-YFP. If the quantification of TCX2-YFP was performed in individual cells and traced over time independent of transcriptome profiling, the number and time point of YFP analyses seems to be too few considering the biological noise.

6. Lines 221-222: Meanwhile, TCX2 expression is predicted to significantly decrease between 4D 16H and 5D in both the Xyl and CSCs, suggesting that these cells divide during this time.

Comment: The anti-correlation between TCX2 expression and cell division need to be shown experimentally before this statement is made. For example, double imaging and quantification can be made in the root expressing both TCX2::TCX2:YFP and WOX5::WOX5:GFP to find whether cells with decreasing TCX2-YFP level also show decrease in WOX5-GFP.

Minor Comments:

7. Supplemental tables are not labeled and very difficult to read.

8. Equations used in the network inference could not be found in the supplemental information.

9. Reference 16 needs to be updated

Reviewer #3 (Remarks to the Author):

This paper takes the interesting approach of seeking to identify genes that are in common between stem cells of different identity, using sorted stem cells from Arabidopsis roots as the model system. Using an exemplary systems approach, these genes were then subjected to gene regulatory network inference, and an Arabidopsis LIN54 homolog was identified as a high scoring gene based on network connectivity measures. LIN54 is a component of the mammalian DREAM complex involved in controlling cell cycle gene expression both in quiescence and during the cell cycle. Further phenotypic analysis confirmed a mild phenotype on stem cell behavior and changes in regulation of predicted target genes, whose dynamics are modeled and tested. The dynamics of TCX2 expression show it is closely linked to stem cell division in different cell types. Recent work (referenced by this manuscript in BioRxiv) has shown a role for the same gene in the division of guard mother cells.

This is a stimulating and well written paper with broad interest, in part arising from the identification of key factors defining the common identity of stem cells (which have largely been studied from a perspective of having unique identifying transcriptional signatures depending on their lineage, rather than shared characteristics), but also because the approach taken represents an elegant fusion of technically advanced transcriptional profiling using cell sorting, GRN inference, network prediction, modeling and phenotypic analysis. I therefore regard this as an influential paper- indeed the type of paper I would choose to use as an exemplar of an integrated systems and biological analysis of an interesting question (i.e. how activity is coordinated between different stem cells).

Specific comments

TCX2 role: It is mentioned that TCX2 is a homolog of LIN54, a component of the DREAM mammalian complex. However, it is not the homolog that has previously been identified through proteomic analysis, although the genes are related. Some additional discussion of this point and the potential role as part of DREAM would be helpful to put this into context, and the potential relevance of DREAM-like complexes in determining stem cell identity and division.

Fig. 1C: arrows are unclear.

Line 289: missing reference.

GRN Inference: I do not think sufficient information is provided here to replicate the analysis (together with the methods in Ref.11). For example, the input for inferring the individual GRNs is not clear to me and I am unclear how the individual GRNs were combined to form the final network. Please provide sufficient information for these steps to be replicated, particularly in view of dependence on (as yet) non-peer reviewed literature.

Similarly insufficient information is given to replicate the Network Motif Score analysis.

Reviewer #4 (Remarks to the Author):

Plants, like multicellular animal systems, contain stem cells to coordinate organ patterning and regeneration. The current manuscript takes advantage of a wealth of marker and transcriptomic data to probe the underlying gene networks operating in Arabidopsis root stem cells.

The authors initially analyse transcriptomes of root stem versus non-stem cells, to identify a set of > 9000 stem cell expressed genes, >69 of which were enriched in 5 out of 6 stem cell populations, versus nearly 50% of genes occurring in only one stem cell type. Hence, each stem cell appears to have its own (largely) unique transcriptome, yet a common small core set of genes are shared.

To probe the regulatory relationships between these classes, the authors employed network inference approach. They conclude, based on cell specific genes being on the outside of the network, that these must be downstream of the cell-ubiquitous genes (which cluster at the core). However, how can they be so sure of this regulatory directionality? Could the cell-type specific genes function to control a common core set of cell-ubiquitous genes (eg cell cycle) as commonly assumed in animal systems? (and later shown in this manuscript to be the case for TCX2).

The authors next employ network motif scoring (NMS) to reveal enrichment of these classes in key network motifs. This revealed higher scores for cell-ubiquitous (rather than cell-type specific) genes, contrary to what is observed and/or assumed in animal systems.

The remaining studies characterised the functional role of a stem-cell ubiquitous regulator, TCX2, which is enriched in 5 out of 6 stem cell types and highly interconnected via NMS weightings.

Functional studies revealed knock down/out lines exhibit faster root growth and, far less clearly, less organised stem cell organisation. The differences between cell-types in WT versus *tex2* mutant roots appears small (in Fig 3A), even when employing stem cell specific markers (Fig 3B). One interesting observation was that cell-type specificity of the markers was 'relaxed' in the *tcx2* background with ectopic expression in selected adjacent stem cell types for several markers.

Intriguingly, TCX2 is predicted to be regulated by, and also targets, cell type-specific genes. This illustrates that the conclusion (based on data in Fig 2A) about stem cell specific and stem-ubiquitous binary regulatory relationship is likely to be too simplistic.

To validate the network described in Fig 3, changes in TCX2 and/or its targets in loss of function or over-expression lines were performed. Whilst this clearly reveals changes in target expression and whether this relationship is activating/repressing, it does not confirm whether TCX2 is a direct target for the identified regulators or whether it, in turn, directly regulates its own set of named targets. I was surprised Chip-PCR or Y1H experiments had not been performed or public datasets (eg SHR Chip-related data or DIP-seq for other TFs) were utilised to address this question. Answers to this point are particularly important given the authors subsequently propose dynamic changes in TCX2 levels are important for regulation of its targets (and stem cell division activity). If these relationships are indirect, this suggests intermediate regulatory mechanisms amplify the impact of small differences in TCX2 levels detected. Subsequent time-course transcriptome analysis (at 8 hour time intervals) does not provide sufficient temporal resolution to infer whether TCX2 directly or indirectly regulates the named targets.

Irrespective, elegant scanning FCS analysis is used to demonstrate that TCX2 does not move between cells, therefore functioning cell-autonomously. Quantification of TCX2-YFP (is this fully functional?) reveals a fascinating correlation between elevated TCX2 abundance and repression of division in the QC, and reduced TEX2 levels are division in other stem cell populations. These latter results are particularly interesting and, I feel, the authors need to make more of this, rather than the relatively routine GRN related results. For example, how does this mitotic repressive role fit with its function as part of the DREAM complex? Hence, more needs to be made of the biological insights!

Reviewers' comments:

Reviewer #1 (Remarks to the Author):

Clarke et al. present a transcriptome analysis of the known stem cell types in the root apical meristem and a comparison to “non-stem cells” of the root. They used these data to define genes that are expressed in many stem cell types (stem cell-ubiquitous genes) and those that are expressed in just one or a few stem cell types (stem cell-specific genes). Gene regulatory network inference was then used to derive gene networks for each cell type, allowing them to ask whether stem-cell ubiquitous factors might regulate genes in a stem cell-specific manner and thus contribute to the activity of the stem cell pool. Analysis of the network characteristics identified TCX2 as a stem cell-ubiquitous factor with many predicted downstream targets and relatively few predicted upstream regulators, which could be expected for a “hub” gene in the network. Genetic analysis showed that *tcx2* mutants indeed had defects in cell division control in most of the root stem cell types, and that some of the genes proposed to be first network neighbors of TCX2 showed alterations in expression in the *tcx2* mutants. Finally, the authors performed a time course transcriptome analysis of the root tip and used these data, along with TCX2 imaging, to generate mathematical models of TCX2 expression and network connections over time in each stem cell

type. This led to the accurate predictions of the timing of each stem cell, based on a reduction in TCX2 levels. Overall, this is an outstanding study that defines new roles for TCX2 and hypothesizes mechanisms for its action in regulating stem cell division. In my opinion, this manuscript needs only minor revisions prior to publication. Below I point out issues that I think should be addressed, in decreasing order of importance:

1. Regarding the TCX2 regulatory network, the authors should at least point out that these are inferred and thus remain to be tested directly (in subsequent studies) by determining TCX2 binding in each cell type as well as by transcriptome profiling of each cell type in WT and *tcx2* mutants. Along this line, it would be nice to see some discussion of why the expression changes of TCX2 first-neighbors are so subtle in the *tcx2* mutant (Extended Data Figure 4B).

We thank the reviewer for this comment. We have better clarified that the predicted regulations in the TCX2 network should be tested directly in subsequent studies in the discussion.

To better validate the TCX2 network, we have completed cell type specific profiling of the QC, Epi/LRC, CEI, and Xyl initials in the *tcx2* mutant. Using these data, we are able to validate that 78% of the predicted TCX2 direct targets have a significant (fold change > 1.5) change in expression in any one of these cell types in the *tcx2* mutant. As seen in Supplemental Table 5, many of the predicted downstream targets have much larger fold changes than observed in the qPCR results in Supplementary Figure 5. We believe this is because qPCR was performed on tissue from the root tip, while our new RNA-Seq data are performed on a cell type specific level. Thus, the fold changes seen in our cell type specific profiling should be a more accurate indication of the degree of activation/reduction of TCX2's predicted direct targets.

Additionally, in an attempt to validate some direct binding of TCX2, we mined a DAPSeq dataset (O'Malley et al, 2016, Cell) performed in the Arabidopsis leaf. Although we did not expect a perfect overlap since our networks are inferred using root data and these DAPSeq data are from the leaf, we were able to validate that 15% of predicted direct targets of TCX2 are directly bound by TCX2. We include this validation in Supplemental Table 4.

2. I also wonder whether the non-stem cell control (NSC; Figure 1B) used in this study was the best choice. This control is the transcriptome of GFP negative cells from the AGL42:GFP sort. Given that AGL42 only appears to be expressed in only 3 of the 6 stem cell types (Fig 1 A and Extended data Fig 1B), the NSC still contains three other stem cell types. Also, the nature of the NSC should be described in the main text so that readers understand what it is.

The image we show of AGL42:GFP in Figure 1A is a representative image. The expression profile of AGL42:GFP is heterogeneous between roots. We provide attached to this response a variety of AGL42:GFP images showing that it is expressed in all of the stem cells, but the degree is different between roots. Additionally, we believe the NSC is a good control because as described in the results the NSC group is very far away from all of the stem cell samples in the PCA. This suggests that the NSC is very biologically different from the stem cells, supporting that we collected a population of cells of which the majority are non-stem cells. We have improved the description of the NSC sample in the results and methods.

3. The timeframe of the “stem cell time course” and associated models and data in Figure 4 need some clarification. The time course is stated as being from 4D to 6D - is this days post-germination? I assume that this actually just represents any two diurnal cycles, since the stem cells are dividing continuously, rather than only once after 4 to 6 days. Is it possible to describe this as a 48 hour time course (e.g. with time zero being at sunrise on day 1) so as to be more intuitive? For example, the statement on Lines 179-181 that “most regulation of TCX occurs between 4-5 days, as does important stem cell divisions” is confusing.

We have included much more detail on the time course within the results and in the methods. Plants were collected from 4 days to 6 days after plating (not post-germination), and plants were collected at the same time of day to eliminate circadian effects.. These data are also now available on GEO, accession # GSE131988

4. In Figure 2A, is it possible to highlight some of the pertinent connections between stem cell ubiquitous and stem cell-specific networks? One can't really see much of what is going on here in that regard due to the high density. Also, there are no visible arrows, bars, or open circles as referenced in the legend.

We thank the reviewer for this comment but we believe highlighting some of the connections will add to the complexity of the figure. The main goal of Figure 2A is to show that the cell ubiquitous genes are towards the center and the cell specific genes are towards the outside of the network. We have removed the details about arrows, etc in the legend as they are not visible. Later, we pull out the subnetwork of TCX2 as an example of how cell ubiquitous and cell specific networks are connected.

5. There are a number of grammatical mistakes in the manuscript that need to be corrected.

We have made an effort to correct these mistakes. Please let us know if you see any additional glaring grammatical errors.

6. The Extended Data tables should be in spreadsheet format so that they can be more easily used by readers.

In this revision, we include the extended data tables in .xlsx format so that they can be viewed in Microsoft Excel or comparable software.

Reviewer #2 (Remarks to the Author):

In the manuscript NCOMMS-18-38553-T titled ‘Stem-cell-ubiquitous genes spatiotemporally coordinate division through regulation of stem-cell-specific gene networks’, authors infer the gene regulatory networks based on the cell type specific transcriptome profiling for major cell type precursors in the Arabidopsis root meristem and find that the predicted core stem cell regulators acting upstream of specific cell type determinants tend to be stem cell ubiquitous. Among those core regulators, authors report that TCX2 coordinates the stem cell division by dynamically changing its expression domain in different cell type precursors.

Overall, the manuscript well integrates the GRN inference and biological validation, which will contribute to advancing the plant stem cell research. However, several key aspects need to be fully addressed to make the manuscript scientifically thorough.

Major comments:

1. Authors conclude that core stem cell regulators in plants tend to be ubiquitous whereas those in animals are cell-type specific (lines 100-101). This is based on the study by D'Alessio et al. (2015), in which authors searched for core regulators that likely determine fates of different cell types in human, and found regulators that conferred retinal pigment epithelial cell identity when they were overexpressed. In my opinion, the work by D'Alessio et al. (2015) was performed in different contexts from this manuscript. D'Alessio et al.'s aimed to find cell type determinants whereas this manuscript aimed to find coordinators of cell type determinants. Thus, contrasting and generalizing cell type expression patterns of stem cell regulators in plants and animals based on these two studies seems an overstatement.

We have expanded the introduction to provide more detail on local mechanisms shown to control stem cell pluripotency in animals and have provided more background on stem cell maintenance in plants in our now expanded Introduction section. We have also softened our statement on the D'Alessio et al paper.

2. This manuscript does not explain clearly about the significance of coordinating the timing of stem cell division in the stem cell maintenance of the root meristem. It would be helpful if authors provide the long-term consequence of perturbing TCX2 or some other relevant data.

We thank the reviewer for this comment. To address it, we have performed long-term imaging on the *tcx2* mutant as well as a TCX2 overexpression line. We see that there continues to be a slight increase in root length in the *tcx2* mutant, although it becomes more subtle as time progresses. For the TCX2 overexpression line, we see plants are much smaller even at later time points, supporting that an increase in TCX2 expression causes less cell division (in contrast with the higher cell division in the *tcx2* mutant). While these results are interesting, we feel that they detract from the main message of our paper, which focuses on *tcx2*'s role in the stem cells in 4-6 day old plants. Therefore, we include these images as an attachment to the reviewer but have decided not to include them in the paper as they do not add any significant conclusions pertaining to our particular biological question.

The images labeled "N35" are of the *tcx2* mutant plated alongside a WT control. Our TCX2 overexpression line (TCX2OX) is beta-estradiol inducible, so images labeled "B" have TCX2 overexpressed while images labeled "MS" are the seedlings plated on control media and have endogenous TCX2 levels.

3. Lines 49-51: the PCA shows that the non-stem cell samples (red) are distant (using the Euclidean distance metric) from all of the stem cell populations, suggesting that the stem cells have a different transcriptional signature than the non-stem cells (Figure 1B).

Comment: It is not clear to me by looking at the PCA plot. It would be helpful if the coordinates of each sample or the Euclidean distance matrix is provided.

The PCA plot is displayed in the principal component space, where each of the 3 axes (x, y, z) represents one of the top 3 components of variance. The way that one interprets a PCA plot is that the further apart samples are along the axes, the larger difference they have in that component of variance. Euclidean distance measures are often not included because the principal component space is a transformation based on the original data, and so the exact distance in the PCA plot only correlates with the difference in the samples and is not an exact measure. We believe that it is clear in Figure 1B that the red group is located very far away from all of the other groups and hope that this explanation helps the reviewer to see what we see. PCA is widely used across biology, so we believe a more detailed explanation of the PCA results is beyond the scope of this paper.

4. From line 175-

The experimental approaches used to infer the time-dependent networks of TCX2 are not clear. Describe the experiments in more detail in materials and methods.

We have expanded the approach for the network inference in the methods. In our methods section titled "Gene Regulatory Network Inference", we detail our network inference method, RTP-STAR, and provide a link to our GitHub repository where code can be viewed. We also clarify what data are used for the network inference for each method.

5. From line 206:

It is not clear to me how authors specified times for time series experiments of transcript profiling and quantification of TCX2-YFP. If the quantification of TCX2-YFP was performed in individual cells and traced over time independent of transcriptome profiling, the number and time point of YFP analyses seems to be too few considering the biological noise.

We thank the reviewer for this comment. First, we clarify that we chose these time points based on our time course data of TCX2 showing that there is a significant decrease of TCX2 between 4D 16H and 5D. We wanted to better quantify the expression of TCX2 in this time frame given that it is the only time interval where its expression significantly changes. We have also increased the number of cells we tracked in the TCX2:TCX2-YFP lines. We now perform YFP tracking in 21 different roots (previously 8 roots) which greatly increases the number of cells tracked and improves our clustering. We have updated the results presented in Supplementary Figure 9 to reflect the higher number of biological replicates.

6. Lines 221-222: Meanwhile, TCX2 expression is predicted to significantly decrease between 4D 16H and 5D in both the Xyl and CSCs, suggesting that these cells divide during this time.

Comment: The anti-correlation between TCX2 expression and cell division need to be shown experimentally before this statement is made. For example, double imaging and quantification can be made in the root expressing both TCX2::TCX2:YFP and WOX5::WOX5:GFP to find whether cells with decreasing TCX2-YFP level also show decrease in WOX5-GFP.

To address the effect of TCX2 expression on cell division, we crossed the cell division marker CYCB1;1:GFP into the *tcx2* mutant and used our temporal tracking pipeline to track CYCB1;1 expression in *tcx2* mutant and WT plants over time.

We find a number of effects of the *tcx2* mutant on CYCB1;1:GFP expression that suggest that TCX2 modulates cell division. First, the average expression per cell of CYCB1;1:GFP is higher in the *tcx2* mutant than in WT. Second, a higher proportion of cells in the *tcx2* mutant have high CYCB1;1:GFP expression (which we define as an intensity value >0.005), while a lower proportion of cells in the *tcx2* mutant have low CYCB1;1 expression (which we define as intensity < 0.0025), supporting that the increase in the mean CYCB1;1:GFP intensity in the *tcx2* mutant is due to more cells with high GFP expression. These two results suggest that CYCB1;1 might be more active in the *tcx2* mutant, which supports the extra division phenotype we see. Finally, we find that significantly less cells have CYCB1;1:GFP expressed for 2 consecutive time points in the *tcx2* mutant compared to WT, suggesting that the CYCB1;1:GFP expression does not persist as long in the *tcx2* mutant. This result suggests that cells in the *tcx2* could be dividing faster than in WT. Taken together, this new analysis supports TCX2's role in cell division and suggests a possible mechanism through modulating CYCB1;1 expression. These new results are presented in Extended Data Fig 4.

Minor Comments:

7. Supplemental tables are not labeled and very difficult to read.

As reviewer #1 suggested we have included all supplemental tables as .xlsx files so that font size/resolution is not an issue and tables are easier to read.

8. Equations used in the network inference could not be found in the supplemental information.

We provide more information on the network inference pipeline in the methods as described in point 4. The core component of RTP-STAR is GENIE3, and equations can be found in the cited reference (Huynh-Thu et al, 2010)

9. Reference 16 needs to be updated

We have fixed reference 16, which is now reference 15.

Reviewer #3 (Remarks to the Author):

This paper takes the interesting approach of seeking to identify genes that are in common between stem cells of different identity, using sorted stem cells from Arabidopsis roots as the model system. Using an exemplary systems approach, these genes were then subjected to gene regulatory network inference, and an Arabidopsis LIN54 homolog was identified as a high scoring gene based on network connectivity measures. LIN54 is a component of the mammalian DREAM complex involved in controlling cell cycle gene expression both in quiescence and during the cell cycle. Further phenotypic analysis confirmed a mild phenotype on stem cell behavior and changes in regulation of predicted target genes, whose dynamics are modeled and tested. The dynamics of TCX2 expression show it is closely linked to stem cell

division in different cell types. Recent work (referenced by this manuscript in BioRxiv) has shown a role for the same gene in the division of guard mother cells.

This is a stimulating and well written paper with broad interest, in part arising from the identification of key factors defining the common identity of stem cells (which have largely been studied from a perspective of having unique identifying transcriptional signatures depending on their lineage, rather than shared characteristics), but also because the approach taken represents an elegant fusion of technically advanced transcriptional profiling using cell sorting, GRN inference, network prediction, modeling and phenotypic analysis. I therefore regard this as an influential paper- indeed the type of paper I would choose to use as an exemplar of an integrated systems and biological analysis of an interesting question (i.e. how activity is coordinated between different stem cells).

Specific comments

TCX2 role: It is mentioned that TCX2 is a homolog of LIN54, a component of the DREAM mammalian complex. However, it is not the homolog that has previously been identified through proteomic analysis, although the genes are related. Some additional discussion of this point and the potential role as part of DREAM would be helpful to put this into context, and the potential relevance of DREAM-like complexes in determining stem cell identity and division.

We thank the reviewer for this comment and agree the role of TCX2 as a potential DREAM homologue is an interesting discussion. We have included some thoughts on this in the discussion section.

Fig. 1C: arrows are unclear.

We are not sure what arrows the reviewer is referring to in Figure 1C.

Line 289: missing reference.

Reference is now included.

GRN Inference: I do not think sufficient information is provided here to replicate the analysis (together with the methods in Ref.11). For example, the input for inferring the individual GRNs is not clear to me and I am unclear how the individual GRNs were combined to form the final network. Please provide sufficient information for these steps to be replicated, particularly in view of dependence on (as yet) non-peer reviewed literature.

Similarly insufficient information is given to replicate the Network Motif Score analysis.

We have expanded our “Gene Regulatory Network Inference” section that details both our network inference pipeline, RTP-STAR, and our network motif score analysis. Our code for RTP-STAR is now provided in a GitHub repository (<https://github.com/nmclark2/RTP-STAR>) along with test data the reviewers can use to test the pipeline. We have also included a new section on the Network Motif Score which details how it is calculated.

Reviewer #4 (Remarks to the Author):

Plants, like multicellular animal systems, contain stem cells to coordinate organ patterning and regeneration. The current manuscript takes advantage of a wealth of marker and transcriptomic data to probe the underlying gene networks operating in Arabidopsis root stem cells.

The authors initially analyse transcriptomes of root stem versus non-stem cells, to identify a set of > 9000 stem cell expressed genes, >69 of which were enriched in 5 out of 6 stem cell populations, versus nearly 50% of genes occurring in only one stem cell type. Hence, each stem cell appears to have its own (largely) unique transcriptome, yet a common small core set of genes are shared.

To probe the regulatory relationships between these classes, the authors employed network inference approach. They conclude, based on cell specific genes being on the outside of the network, that these must be downstream of the cell-ubiquitous genes (which cluster at the core). However, how can they be so sure of this regulatory directionality? Could the cell-type specific genes function to control a common core set of cell-ubiquitous genes (eg cell cycle) as commonly assumed in animal systems? (and later shown in this manuscript to be the case for TCX2).

We thank the reviewer for this comment. The organic network layout chosen in Figure 2A puts the top of the network in the center, and the bottom of the network at the edges. We agree it is interesting to think about how cell-ubiquitous and cell-specific genes control each other on a global scale, but we think it is beyond the scope of this paper. Here, we focus on one example of how a cell-ubiquitous gene controls aspects of cell division and identity through regulation of cell-specific targets, but we hope that in the future this connection between cell-specific and cell-ubiquitous genes is further investigated. We have highlighted this topic in the discussion section of our manuscript.

The authors next employ network motif scoring (NMS) to reveal enrichment of these classes in key network motifs. This revealed higher scores for cell-ubiquitous (rather than cell-type specific) genes, contrary to what is observed and/or assumed in animal systems.

The remaining studies characterised the functional role of a stem-cell ubiquitous regulator, TCX2, which is enriched in 5 out of 6 stem cell types and highly interconnected via NMS weightings.

Functional studies revealed knock down/out lines exhibit faster root growth and, far less clearly, less organised stem cell organisation. The differences between cell-types in WT versus *tex2* mutant roots appears small (in Fig 3A), even when employing stem cell specific markers (Fig 3B). One interesting observation was that cell-type specificity of the markers was 'relaxed' in the *tcx2* background with ectopic expression in selected adjacent stem cell types for several markers.

To further show the large effect of TCX2 on cell division and identity, we completed cell type specific sorting of the QC, Epi/LRC, CEI, and Xyl initials in the *tcx2* mutant. We find that a large number of genes (3000 to 10000 depending on the cell type) are differentially expressed in the mutant compared to the WT, and that this number is a similar magnitude as the number of

genes found to be enriched in these cell types. We feel that this additional transcriptional profiling makes up for the perhaps subtle phenotype seen in the roots. Results from this are shown in Supplemental Table 5

Intriguingly, TCX2 is predicted to be regulated by, and also targets, cell type-specific genes. This illustrates that the conclusion (based on data in Fig 2A) about stem cell specific and stem-ubiquitous binary regulatory relationship is likely to be too simplistic.

To validate the network described in Fig 3, changes in TCX2 and/or its targets in loss of function or over-expression lines were performed. Whilst this clearly reveals changes in target expression and whether this relationship is activating/repressing, it does not confirm whether TCX2 is a direct target for the identified regulators or whether it, in turn, directly regulates its own set of named targets. I was surprised Chip-PCR or Y1H experiments had not been performed or public datasets (eg SHR Chip-related data or DIP-seq for other TFs) were utilised to address this question. Answers to this point are particularly important given the authors subsequently propose dynamic changes in TCX2 levels are important for regulation of its targets (and stem cell division activity). If these relationships are indirect, this suggests intermediate regulatory mechanisms amplify the impact of small differences in TCX2 levels detected. Subsequent time-course transcriptome analysis (at 8 hour time intervals) does not provide sufficient temporal resolution to infer whether TCX2 directly or indirectly regulates the named targets.

Please see our response to reviewer #1, point #1 to see extra steps we have taken to better validate direct regulation of TCX2 on its direct targets. While our validation is a good initial step, we agree that more work needs to be done to validate direct regulation of TCX2 on its targets as the DAPSeq data we use are in the leaf. However, we agree with reviewer #1 that this type of analysis is beyond the scope of this work and should be included in future studies. We further expand on this in the discussion section of our manuscript.

Irrespective, elegant scanning FCS analysis is used to demonstrate that TCX2 does not move between cells, therefore functioning cell-autonomously. Quantification of TCX2-YFP (is this fully functional?) reveals a fascinating correlation between elevated TCX2 abundance and repression of division in the QC, and reduced TEX2 levels are division in other stem cell populations. These latter results are particularly interesting and, I feel, the authors need to make more of this, rather than the relatively routine GRN related results. For example, how does this mitotic repressive role fit with its function as part of the DREAM complex? Hence, more needs to be made of the biological insights!

The TCX2:TCX2-YFP line (also called SOL2:SOL2-YFP) is shown to fully complement the mutant phenotype in stomata in Simmons et al, 2019, Development, and we observe complementation of the mutant phenotype in roots, so it is fully functional. To better correlate TCX2 expression with changes in cell division, we refer the reviewer to our response to reviewer #2, point #2. As to how this may fit with its function as part of the DREAM complex we refer the reviewer to our response to reviewer #3, point #1.

To specifically address TCX2 repression of QC division, we transcriptionally profiled cells expressing WOX5:GFP in *tcx2* mutants, which should be mostly QC cells (although, since the WOX5:GFP marker expands into the CEI in the *tcx2* mutant as shown in Figure 3, it is possible some of these genes are expressed in the CEI), and found 3024 genes are differentially expressed compared to WT (Supplemental Table 5). We find this list of DE genes contains a number of known cell cycle related proteins such as CYCA1;2; CYCA2;4; CYCD3;2; CYCLIN DEPENDENT KINASE GROUP C2 (CDKC2); and CDKB1;1, which are all core cell cycle genes; SIAMESE (SIM) which contains a cyclin binding motif and is known to coordinate cell division in trichomes; KIP-RELATED PROTEIN 5 (KRP5) which encodes a cyclin-dependent kinase inhibitor and is a known negative regulator of cell division; EMBRYO DEFECTIVE 2769 (EMB2769) which is a cell cycle control family protein; CELL CYCLE SWITCH PROTEIN 52 B (CCS52B) which has been implicated in root hair and trichome division; HIGH ARSENIC CONTENT 1 (HAC1) which is a cell cycle control phosphatase; KINETOCHORE NULL 2 (KNL2) which localizes to centromeres during mitosis; RIBONUCLEOTIDE REDUCTASE 2A (RNR2A) which is critical for cell cycle progression; and others. We believe this transcriptional profiling helps strengthen our conclusion that TCX2 regulates QC division.

Additionally, we performed quantification of QC division and WOX5:GFP fluorescence in the *tcx2* mutant vs WT at three different time points (4D 16H, 5D 16H, 6D 16H) to look at the effects of the *tcx2* mutant over a longer period of time. At all of the time points we see a significant decrease in WOX5:GFP expression in the QC, which supports our quantification results at 5D. Further, we find more QC divisions in the *tcx2* mutant plants compared to WT plants at all time points. Although this difference is not statistically significant, it does support our hypothesis that TCX2 represses QC divisions as *tcx2* mutants have more QC divisions. Given that we already present these data in Figure 3A for 5DO roots, we believe this time course does not add much to the conclusions of the paper. However, we include these results in response to the reviewer to show that the trend remains over time.

Corrected Total Cell Fluorescence (CTCF) of WOX5:GFP in *tcx2* mutant (orange) or WT (blue) from $n > 21$ QC cells.

Percent of QC cells in WT or *tcx2* mutant with division (red) or without division (blue) from $n > 21$ QC cells. Error bars are SEM.

REVIEWERS' COMMENTS:

Reviewer #1 (Remarks to the Author):

The authors have addressed all of my concerns about the previous version of the manuscript.

Reviewer #2 (Remarks to the Author):

Remarks to the author:

In the revised manuscript, authors addressed the most of my comments. This manuscript convincingly shows that TCX2 is under the regulation of multiple cell type specific stem cell regulators and in turn it coordinates the spatio- and temporal action of cell type specific stem cell regulators. The dynamic action of TCX2 is also indicated by networks inferred from stem cell transcriptome profiling in time course. Overall, I think this manuscript serves as a nice study case for how dynamic regulatory networks can explain the coordination of cell divisions for organ growth and cell type specific processes.

One unresolved concern, however, is related to the *tcx2* mutant. Its phenotype is so subtle that it is difficult to understand the developmental significance of TCX2-mediated coordination of cell division for proper cell type patterning and root growth. In the time course profiling of stem cell population, TCX2 expression level drops at 5D. What does this mean to the cell division coordination? Does this mean that the dynamic expression of TCX2 occurs periodically to ensure the robustness in the division for cell type patterning and growth? If so, I expect the enhanced disruption in root growth and radial patterning over time. However, this does not seem to be the case. I think authors need to discuss more about the role of TCX2 in coordination of global and local cell division processes, maybe in connection with other TCX members involved in DREAM complexes.

Reviewer #3 (Remarks to the Author):

The authors have carried out comprehensive revision of the manuscript and conducted a number of further analyses to further support the conclusions. In particular they have profiled cell type specific expression in the *tcx2* mutant, further confirming their data and showing higher levels of regulation of putative TCX2 target genes. In my opinion, the revised manuscripts satisfactorily addresses all comments raised.

Reviewer #4 (Remarks to the Author):

The authors have performed an excellent job addressing my own and other reviewers comments. They have also added several additional data sets that support their original conclusions. I am happy with the revisions which have significantly improved the manuscript.

REVIEWERS' COMMENTS:

Reviewer #1 (Remarks to the Author):

The authors have addressed all of my concerns about the previous version of the manuscript.

Reviewer #2 (Remarks to the Author):

Remarks to the author:

In the revised manuscript, authors addressed the most of my comments. This manuscript convincingly shows that TCX2 is under the regulation of multiple cell type specific stem cell regulators and in turn it coordinates the spatio- and temporal action of cell type specific stem cell regulators. The dynamic action of TCX2 is also indicated by networks inferred from stem cell transcriptome profiling in time course. Overall, I think this manuscript serves as a nice study case for how dynamic regulatory networks can explain the coordination of cell divisions for organ growth and cell type specific processes.

One unresolved concern, however, is related to the *tcx2* mutant. Its phenotype is so subtle that it is difficult to understand the developmental significance of TCX2-mediated coordination of cell division for proper cell type patterning and root growth. In the time course profiling of stem cell population, TCX2 expression level drops at 5D. What does this mean to the cell division coordination? Does this mean that the dynamic expression of TCX2 occurs periodically to ensure the robustness in the division for cell type patterning and growth? If so, I expect the enhanced disruption in root growth and radial patterning over time. However, this does not seem to be the case. I think authors need to discuss more about the role of TCX2 in coordination of global and local cell division processes, maybe in connection with other TCX members involved in DREAM complexes.

Thank you for asking for these clarifications. The subtle phenotype of the *tcx2* mutant suggests that there may be some redundancy either among the components of the DREAM complex itself or family-related genes (such as SOL1 as shown in the Arabidopsis stomatal lineage).

To address the question of the potential role of TCX2 in the DREAM complex, we have added additional details in the Discussion. We highlight what is known about the DREAM complex in animals as well as the role of TCX2 and its homologues in plants (lines 323-334).

The reviewer raises an excellent point in regard of the role of TCX2 in global vs local cell division processes, as this remains an open question. While the results in the stomata support that TCX2 acts locally, we cannot say whether there might be a larger global mechanism in stem cells as our network analysis predicts. Consequently, we still leave this as an open-ended question to pursue in future studies.

Reviewer #3 (Remarks to the Author):

The authors have carried out comprehensive revision of the manuscript and conducted a number of further analyses to further support the conclusions. In particular they have profiled

cell type specific expression in the *tcx2* mutant, further confirming their data and showing higher levels of regulation of putative TCX2 target genes. In my opinion, the revised manuscripts satisfactorily addresses all comments raised.

Reviewer #4 (Remarks to the Author):

The authors have performed an excellent job addressing my own and other reviewers comments. They have also added several additional data sets that support their original conclusions. I am happy with the revisions which have significantly improved the manuscript.